



## Evaluation of Lacustrine Groundwater Discharge, Hydrologic Partitioning, and Nutrient Budgets in a Proglacial Lake in Qinghai-Tibet Plateau: Using [222]Rn and Stable Isotopes

**Xin LUO[1, 2], Xing Xing Kuang[3], Jiu Jimmy Jiao[1, 2*], Sihai Liang[4], Rong Mao[1, 3],**
**Xiaolang Zhang[1, 3], and Hailong Li[3]**
[1]Department of Earth Sciences, The University of Hong Kong, P. R. China
[2]The University of Hong Kong, Shenzhen Research Institute (SRI), Shenzhen, P. R.
China
[3]School of Environmental Science and Engineering, South University of Science and
Technology of China (SUSTC), Shenzhen, China.
[4]School of Water Resources & Environment, China University of Geosciences, 29
Xueyuan Road, Beijing, China
Corresponding author: Jiu Jimmy Jiao (jjiao@hku.hk)
Department of Earth Sciences, The University of Hong Kong
Room 302, James Lee Science Building, Pokfulam Road, Hong Kong
Tel (852) 2857 8246; Fax (852) 2517 6912



**Abstract**
Proglacial lakes are good natural laboratories to investigate groundwater and
glacier dynamics under current climate condition and to explore primary productivity
under pristine lake status. This study conducted a series of investigations of $^{222}$Rn,
stable isotopes, nutrients and other hydrogeochemical parameters in Ximen Co Lake,
a remote proglacial lake in the east of Qinghai-Tibet Plateau (QTP). A radon mass
balance model was used to quantify the lacustrine groundwater discharge (LGD) of
the lake, leading to an LGD estimate of $10.3 \pm 8.2$ mm d$^{-1}$. Based on the three end
member models of stable $^{18}$O and Cl$^{-}$, the hydrologic partitioning of the lake is
obtained, which shows that groundwater discharge only accounts for 7.0 % of the
total water input. The groundwater derived DIN and DIP loadings constitute 42.9 %
and 5.5 % of the total nutrient loading to the lakes, indicating the significance of LGD
in delivering disproportionate DIN into the lake. The primary productivity of the lake
water is calculated to be 0.41 mmol C m$^{-2}$ d$^{-1}$. This study presents the first attempts to
evaluate the LGD and hydrologic partitioning in the glacial lake by coupling
radioactive and stable isotopic approaches and the findings advance the understanding
of nutrient budgets and primary productivity in the proglacial lakes of QTP. The study
is also instructional in revealing the hydrogeochemical processes in proglacial lakes
elsewhere.



**Keywords:** Proglacial lake; [222]Rn; lacustrine groundwater discharge; hydrologic

partitioning; nutrient budgets; primary productivity

## 1. Introduction

High altitude and latitude areas are intensively influenced by the melting of

glaciers due to climatic warming. Of particular importance are the proglacial areas,

such as proglacial lakes and moraines, because they are particularly affected by

climatic change induced glacier retreating and thawing of permafrost (Barry 2006,

Heckmann et al. 2015, Slaymaker 2011). The proglacial lakes are usually located

close to ice front of a glacier, ice cap or ice sheet, with the vicinity to the ice front

sometimes defined as the areas with subrecent moraines and formed by the last

significant glacier advances at the end of the Little Ice Age (Barry 2006, Harris et al.

2009, Heckmann et al. 2015, Slaymaker 2011). Proglacial lakes are located in the

transition zones from glacial to non-glacial conditions, providing natural laboratories

to explore hydrological processes, biogeochemical cycles and geomorphic dynamics

under current climatic conditions (Dimova et al. 2015, Heckmann et al. 2015).

Mountainous proglacial lakes, formed by glacial erosion and filled by melting glaciers,

are widely distributed in the Qinghai-Tibet Plateau (QTP), especially along the

substantial glacier retreating areas of Himalaya Mountains (MT.), Qilian MT.,



Tienshan MT., etc. Characterized by higher elevations, small surface areas but
relatively large depths, mountainous proglacial lakes in QTP lack systematic
field-based hydrological studies due to their remote locations and difficulty in
conducting field work (Bolch et al. 2012, Farinotti et al. 2015, Yao et al. 2012).

There has been extensive recognition of the importance of groundwater discharge

to various aquatic systems for decades (Dimova and Burnett 2011, Johannes 1980,
Valiela et al. 1978). Very recently, the topic of 'lacustrine groundwater discharge
(LGD)', which is comprehensively defined as groundwater exfiltration from lake
shore aquifers to lakes (Blume et al. 2013, Lewandowski et al. 2015, Lewandowski et
al. 2013, Rosenberry et al. 2015), has been introduced. LGD is analogous of in
submarine groundwater discharge (SGD) in coastal environments. LGD plays a vital
role in lake hydrologic partitioning, which is defined as the separation of groundwater
discharge/exfiltration, riverine inflow, riverine outflow infiltration, surface
evaporation and precipitation for the hydrological cycle of the lake (Good et al., 2015).
LGD also serves as an importance component in delivering solutes to lakes since
groundwater is usually concentrated in nutrients, $CH_4$, dissolved inorganic/organic
carbon (DIC/DOC) and other geochemical components (Belanger et al. 1985, Dimova
et al. 2015, Lecher et al. 2015, Paytan et al. 2015). Nutrients and carbon loading from
groundwater greatly influences ratios of dissolved inorganic nitrogen (DIN) to



dissolved inorganic phosphate (DIP) (referred as N: P ratios thereafter), ecosystem
structure and the primary productivity of the lake aquatic system (Belanger et al. 1985,
Hagerthey and Kerfoot 1998, Nakayama and Watanabe 2008).
LGD studies utilize various methods including direct seepage meters (Lee 1977,
Shaw and Prepas 1990), geo-tracers such as radionuclides, stable $^2$H and $^{18}$O isotopes
(Gat 1995, Kluge et al. 2007, Kraemer 2005, Lazar et al. 2008), heat and temperature
(Liu et al. 2015, Sebok et al. 2013), numerical modeling (Smerdon et al. 2007, Winter
1999, Zlotnik et al. 2009, Zlotnik et al. 2010) and remote sensing (Anderson et al.
2013, Lewandowski et al. 2013, Wilson and Rocha 2016). Recently, some researchers
started to investigate groundwater dynamics in peri- and proglacial areas, mostly
based on the approaches of numerical modeling (Andermann et al. 2012, Lemieux et
al. 2008a, Lemieux et al. 2008b, Lemieux et al. 2008c, Scheidegger and Bense 2014).
However, the quantification of groundwater and surface water exchange in proglacial
lakes is still challenging due to limited hydrogeological data and extremely seasonal
variability of aquifer permeability (Callegary et al. 2013, Dimova et al. 2015, Xin et al.

2013).

$^{222}$Rn, a naturally occurring inert gas nuclide highly concentrated in groundwater,
can be more applicable in fresh aquatic systems and has been widely used as a tracer
to quantify groundwater discharge in fresh water lakes (Corbett et al. 1997, Dimova et





al. 2015, Dimova and Burnett 2011, Dimova et al. 2013, Kluge et al. 2007, Kluge et al.
2012, Luo et al. 2016, Schmidt et al. 2010) and terrestrial rivers and streams
(Batlle-Aguilar et al. 2014, Burnett et al. 2010, Cook et al. 2006, Cook et al. 2003). Of
particular interest are investigations of temporal $^{222}$Rn distribution in lakes, since it
can be used to quantify groundwater discharge and reflect the locally climatological
dynamics (Dimova and Burnett 2011, Luo et al. 2016). Temporal radon variations
give high resolution estimates of groundwater discharge to lakes over diel cycles,
allowing evaluation of LGD and the associated chemical loadings. However, there has
been no study of radon-based groundwater discharge in mountainous proglacial lakes,
especially for those lakes in the QTP.
This study aims to investigate the groundwater surface water interactions for the
proglacial lake of Ximen Co, by estimating the LGD and evaluating the hydrologic
partitioning of the lake. LGD is estimated with $^{222}$Rn mass balance model, and the
hydrologic partitioning of the lake is obtained with the three endmember model
coupling the mass balance of water, stable isotopes and Cl$^{-}$. Then, LGD derived
nutrients are estimated and the nutrient budgets of the lake are depicted. Finally,
primary productivity of the lake water is calculated based on the nutrient budgets.
This study, to our knowledge, makes the first attempt to quantify the LGD,
hydrologic partition, and groundwater borne nutrients of the proglacial lake in QTP



and elsewhere via the approach integrating multiple tracers. This study provides
insights of hydrologic partitioning in a typical mountainous proglacial lake under
current climate condition and reveals groundwater borne chemical loadings in this
proglacial lake in QTP and elsewhere.

**2.    Methodology**
2.1 Site descriptions
The Nianbaoyeze MT., located at the eastern margin of the QTP and being the
easternmost part of NW-SW trending Bayan Har Shan, is situated at the main water
divide of the upper reaches of Yellow River and Yangtze River (Figure 1). With a peak
elevation of 5369 m, the mountain rises about 500-800 m above the surrounding
peneplain and displays typical Pleistocene glacial landscapes such as moraines,
U-shaped valleys and cirques (Lehmkuhl 1998, Schlutz and Lehmkuhl 2009,
Wischnewski et al. 2014). The present snow line is estimated to be at an elevation of
5100 m (some updated references) (Schlutz and Lehmkuhl 2009). Controlled by the
South Asia and East Asia monsoons, the mountain has an annual precipitation of 975
mm in the southern part and 582 mm in the northwestern part, with 80 % occurring
during May and October (Yuan et al. 2014, Zhang and Mischke 2009). The average
temperature gradient is about 0.55 $^{\circ}$C per 100 m, and the closest weather station,



locating in Jiuzhi town (N: 33.424614°, E: 101.485998) at the lower plains of the
mountain, recorded a mean annual temperature of 0.1 $^{\circ}$C. Snowfalls occur in nearly
10 months of the entire year and there is no free-frost all year around (Böhner 1996,
2006, Schlutz and Lehmkuhl 2009). The precipitation, daily bin-averaged wind speed
and temperature in Aug, 2015 were recorded to be 90 mm, .7 m s$^{-1}$ and 9.5 $^{\circ}$C from
Jiuzhi weather station (Figure 2). The water surface evaporation was recorded to be
1429.8 mm in 2015 from Jiuzhi weather station.
Among the numerous proglacial lakes developed in the U-shaped valleys of the
Nianbaoyeze MT., Ximen Co lake is located at the northern margin of the mountain
with an elevation of 4030 m asl, and is well studied and easily accessible (Lehmkuhl
1998, Schlutz and Lehmkuhl 2009, Yuan et al. 2014, Zhang and Mischke 2009). The
lake was formed in a deep, glacially eroded basin with a catchment area of 50 km$^2$,
and has a mean and a maximum depth of 40 m and 63.2 m, and a surface area of 3.6
km$^2$. The vegetation around the lake is dominated by pine meadows with dwarf shrubs,
rosette plants and alpine cushion (Schlutz and Lehmkuhl 2009, Yuan et al. 2014,
Zhang and Mischke 2009). Mostly recharged by the glacial and snowpack melting
water and regional precipitation, the lake is stratified with an epilimnion depth about
4.4 m in the summer time. The lake is usually covered by ice in the winter time
(Zhang and Mischke 2009). The superficial layer within the U-shaped valley is





characterized by peat, clay and fluvial gravels with a depth about 1-3.5 m.
Discontinuous and isolated permafrost is present at the slope of the valley above the
elevation of about 4150 m. The maximum frozen depth is about 1.5 m for the seasonal
frozen ground around the lake. The seasonal frozen ground serves as an unconfined
aquifer during the unfrozen months from July to October, and groundwater discharges
into the epilimnion of the lake (Schlutz and Lehmkuhl 2009, Wang 1997, Zhang and
Mischke 2009).

2.2 Sampling and field analysis
The field campaign to Ximen Co Lake was conducted from August, 2015, when it is
warm enough to take the water samples of different origins as the studied site is
seasonally frozen. A $^{222}$Rn continuous monitoring station was setup at the southeast
part of the lake, which is fairly flat for setting up our tent and monitoring system.
Surface water samples were collected around the lake, rivers at the upstream and
downstream. Porewater samples were collected at one side of the lake as the other
side is steep and rocky. The basic water quality parameters of conductivity (EC),
dissolved oxygen (DO), TDS, ORP, pH in the water were recorded with the
multi-parameter meter (HANNA, Co.). Relative humidity was recorded with a
portable thermo-hydrometer (KTH-2, Co.). Lake water samples were taken with a





peristaltic pump into 2.5 L glass bottles for $^{222}$Rn measurement with the Big Bottle
system (Durridge, Co.). Surface water samples were filtered with 0.45 µm filters
(Advantec, Co.) in situ and taken into 5 ml, 15 ml, 15 ml and 50 ml Nalgene
centrifugation tubes for stable isotope, major anion, cation and nutrient analysis.
Porewater samples were taken from the lakes shore aquifers with a push point sampler
(M.H.E, Co.) connected to peristaltic pump (Solinst, Co.) (Luo et al. 2014, Luo et al.
2016). 100 ml raw surface water or porewater was titrated with 0.1 µM $H_2SO_4$
cartridge (Hach, Co.) in situ to measure total alkalinity (Hasler et al. 2016, Warner et
al. 2013, White et al. 2016). Porewater was filtered with 0.45 µm syringe filters
(Advantec, Co.) in situ and taken into 5 ml, 15 ml, 15 ml and 50 ml Nalgene
centrifugation tubes for stable isotope, major anion, cation and nutrient analysis. 250
ml porewater was taken for $^{222}$Rn measurement with RAD7 $H_2O$ (Durridge, Co.)
Samples for major cation analysis were acidified with distilled $HNO_3$ immediately
after the sampling.

$^{222}$Rn continuous monitoring station was set up at the northwest of the lake, close

to the downstream of the lake (Figure 1b). Lake water (about 0.5 m) was pumped with
a DC pump (12 V) driven by lithium batteries (100 Ah) and sprinkled into the
chamber of RAD7 AQUA with a flow rate > 2 min $L^{-1}$, where $^{222}$Rn in water vapor
was equilibrated with the air $^{222}$Rn. The vapor in the chamber was delivered into two





large dry units (Drierite, Co) to remove the moisture and circulated into RAD7
monitor, where $^{222}$Rn activities were recorded every 5 mins. A temperature probe
(HOBO$^{@}$) was insert into the chamber to record the temperature of the water vapor.
The monitoring was performed from 11: 31 am, Aug 22$^{nd}$ to 6: 30 am, Aug 24$^{th}$, 2015.
During the period of 1:50-4:30 pm on Aug 22$^{nd}$, a sudden blizzard occurred, leading
to an hourly precipitation about 0.6 mm to the lake area. Daily and hourly
climatological data such as wind speed, air temperature and precipitation were
retrieved from the nearest weather station in Jiuzhi town (N: 33.424614°, E:
101.485998). Water level and temperature fluctuations were recorded with a
conductivity-temperature-depth diver (Schlumberger, Co.) fixed at about 20 cm below
the lake surface and calibrated with local atmospheric pressure recorded by a
baro-diver (Schlumberger, Co.) above the lake. To correct for dissolved $^{226}$Ra
supported $^{222}$Rn, one radium sample was extracted from 100 L lake water with $MnO_2$
fiber as described elsewhere (Luo et al. 2014, Moore 1976).

2.3 Chemical analysis
Major ions were measured with ICS-1100 (Dionex. Co.) in the Department of Earth
Sciences, the University of Hong Kong. The uncertainties of the measurements are
less than 5 %. Nutrients, DIN and DIP were analyzed with flow injection analysis



equipped with auto-sampler (Lachat. Co.) in the School of Biological Sciences, the
University of Hong Kong. Stable $^{18}O$ and $^{2}H$ isotopes were measured with
MOA-ICOS laser absorption spectrometer (Los Gatos Research (LGR) Triple Isotope
Water Analyzer (TIWA-45EP)) at State Key Laboratory of Marine Geology, Tongji
University, Shanghai. The stable isotopic standards and the recovery test has been
fully described elsewhere (Luo et al., 2017). The measurement uncertainty is better
than 0.1 % for $^{18}O$ and 0.5 % for $^{2}H$. $^{226}Ra$ was detected with RAD7 with the method
described elsewhere (Kim et al. 2001, Lee et al. 2012)

2.4 Radon transient model

Previous studies employed a steady state radon-222 mass balance model to

quantify LGD to lentic system such as lakes and wetlands (Dimova and Burnett 2011,
Luo et al. 2016). This model assumes that radon input derived from groundwater
inflow, diffusion and river inflow are balanced by the radon losses of atmospheric
evasion, decay and river outflow. However, recently studies revealed that the steady
state is mainly reached after 2-15 days of constant metrological conditions, and
mostly lentic system can be not be treated as steady state due to rapid radon-222
degassing to the atmosphere driven by wind-induced turbulence (Gilfedder et al.,
2015; Dimova and Burnett, 2011).





Ximen Co lake is demonstrated to be highly stratified with an epilimnion of 4.4
m (Zhang and Mischke 2009). The lake was formed by glacier erosion and the
lakebed is characterized by granite bedrock with a thin sedimentary clay layer.
Previous studies have indicated that sediment with a thickness of 0.7-3.3 m has been
developed on the bedrock and forms the lake shore aquifer, which consists of clay,
soils and gravels (Schlutz and Lehmkuhl 2009). Porewater sampled in the aquifer
immediately behind the lake shore can well represent groundwater discharging into
the lake, as suggested previously (Lewandowski et al. 2015, Rosenberry et al. 2015,
Schafran and Driscoll 1993). LGD has been widely considered to occur within the
first few meters of the lake shore (Lee et al. 1980, Rosenberry et al. 2015, Schafran
and Driscoll 1993) and groundwater is considered to predominately discharge into the
epilimnion since deep groundwater flow is highly limited by the Precambrian bedrock
(Einarsdottir et al., 2016). Therefore, [222]Rn mass balance model is established to
quantify LGD to the epilimnion from the lake shore. Due to negligible hydrological
connection between the epilimnion and hypolimnion, LGD for the lake can be
quantified with [222]Rn mass balance model for the epilimnion.
The governing equation of radon-222 transient mass balance model within a 1 x
1 x z cm (where z is the depth in cm) can be expressed as (Gilfedder et al. 2015):
$$z \frac{\partial I_w}{\partial t} = F_{gw} + (I_{226_{Ra}} - I_w) \times z \times \lambda_{222} + F_{diff} - F_{atm} \qquad (1)$$





where $F_{gw}$, $F_{diff}$, $F_{atm}$ [Bq m$^{-2}$ d$^{-1}$] are $^{222}$Rn loadings from LGD, water-sediment
diffusion and water-air evasion, respectively; $z$ [m] is the lake water level depth
recorded by the diver. $\lambda_{222}$ is the decay constant of $^{222}$Rn with a value of 0.186 d$^{-1}$.
$\lambda_{222} \times I_{^{226}Ra}$ and $\lambda_{222} \times I_w$ account for the production and decay of $^{222}$Rn [Bq m$^{-2}$ d$^{-1}$]
in the water column, respectively. $I_w$ and $I_{^{226}Ra}$ [Bq m$^{-2}$] represent $^{222}$Rn and $^{226}$Ra
inventories in the epilimnion, and are expressed as: $I_w = H \times C_w$ and
$I_{^{226}Ra} = H \times C_{^{226}Ra}$, respectively; where $H$ [m] is the depth of the epilimnion; $C_w$ and
$C_{^{226}Ra}$ is the $^{222}$Rn and $^{226}$Ra activity [Bq m$^{-3}$], respectively.
The model is valid under the following assumptions: 1) The epilimnion is well
mixed which is the actual condition for most natural boreal and high altitude glacial
lakes (Åberg et al. 2010, Zhang and Mischke 2009). 2) $^{222}$Rn input from riverine
water inflow, and loss from the lake water outflow and infiltration into the lake shore
aquifer is negligible compared to the groundwater borne $^{222}$Rn, because $^{222}$Rn
concentration of groundwater is 2-3 orders of magnitude larger than that lake water
(Dimova and Burnett 2011, Dimova et al. 2013). Generally, $^{222}$Rn in the epilimnion is
sourced from LGD and decay input from parent isotope of $^{226}$Ra under secular
equilibrium, and is mainly lost via atmospheric evasion and radioactive decay.
$F_{atm}$ is the key sinking component of the transient model and is finally a function of
wind speed and water temperature, both of which are temporal variant variables





(Supplementary information). Lake water level $z$ is also a temporal variant variable
which represents the fluctuations of water volume of the epilimnion. This equation is
discretized by the forward finite difference method, and the groundwater flux at each
time step can be solved as follow
$$[^{222}Rn_{t+\Delta t}] = \frac{[z \times {}^{222}Rn_t + [F_{diff} + F_{gw} - F_{atm} - {}^{222}Rn_t \times \lambda \times z] \times \Delta t}{z}$$      (2)
where ${}^{222}Rn_{t+\Delta t}$ and ${}^{222}Rn_{t+\Delta t}$ [Bq m$^{-3}$] is the $^{222}$Rn activity at current time step and at
the previous time steps, respectively, and $\Delta t$ [min] is the time step. With the inverse
calculation based on Equation (4), the groundwater inflow at each time step can be
obtained. However, large errors of the final LGD calculation will be induced by even
a small amount of noise in the measured $^{222}$Rn data due to the ${}^{222}Rn_{t+\Delta t} - {}^{222}Rn_t$ term
being with the measure uncertainty. To reduce the random errors of the
measured $^{222}$Rn concentrations, the time window with a width of 1 hour is proposed to
smooth the curve (Supplementary information).

3. **Results**
3.1 Time series data
Figure 2 shows the basic climatological parameters of the lake catchment during
the campaign month. There are discrete rainfall events occurring throughout the
month with an average rainfall of 3.1 mm d$^{-1}$. The temperature throughout the month
ranges from 5.0 – 12.5 $^{o}$C within an average of 9.3 $^{o}$C. The daily averaged wind speed



generally ranges from 0.7 – 2.5 m s$^{-1}$, with an average of 1.7 m s$^{-1}$. $^{222}$Rn temporal
distribution and other time series data are shown in Figure 3a and listed in
Supplementary Table 1. Generally, $^{222}$Rn concentration varies from 32.2 to 273 Bq m$^{-3}$,
with an average of 144.2 ± 27.7 Bq m$^{-3}$. $^{222}$Rn over the monitoring period shows
typical diel cycle, much higher at nighttime and lower in the day time. Figures 3b-3d
shows the time series data of temperature (5 mins interval), nearshore lake water level
(1 min interval), and wind speed (1 hour interval). Temperature and lake water level
also show typical diel cycles, but with antiphase fluctuations with each other.
Temperature is higher during the daytime and lower at nighttime. However a sudden
decrease of temperature was recorded due to the sudden blizzard (Figure 3b). Water
level is higher at nighttime and lower during the daytime, with a strong fluctuation
due to the turbulence caused by the blizzard (Figure 3c). The variability might reflect
the dynamics of groundwater input and surface water inflow. The air temperature of
the lake area is in phase with the water temperature. Wind speed is normally higher
during the daytime and lower at nighttime (Figure 3d).
The variation of $^{222}$Rn is nearly in antiphase with the fluctuations of lake water
temperature and air temperature, indicating that the dominated controlling factors
of $^{222}$Rn fluctuations are water temperature and wind speed (Figure 3a). This
phenomenon is reasonable as lake water $^{222}$Rn is predominately lost via atmospheric



evasion, which is the function of wind speed and water temperature (Dimova et al.
2015, Dimova and Burnett 2011, Dimova et al. 2013). High water temperature and
wind speed leads to elevated atmospheric evasion and causes the decline of $^{222}$Rn
concentration in the lake water. However, there is a sudden reduction of radon activity
from 2: 00 pm to 4: 00 pm on Jul 22$^{nd}$, 2015, when the snow event led to a sudden
decrease of water temperature, increase of wind speed, and large surface water
turbulence as indicated by water level fluctuations (Figures 3a-3d). $^{222}$Rn in the
porewater is 2-3 orders of magnitude larger than $^{222}$Rn in the lake water, suggesting
that $^{222}$Rn is an ideal tracer to estimate the LGD (Supplementary Table 1). $^{222}$Rn
concentrations in surface water range from 22.2 to 209 Bq m$^{-3}$, with an average of
92.5 Bq m$^{-3}$ (n = 12), which is in the range of $^{222}$Rn continuous monitoring results,
suggesting reliable $^{222}$Rn measurements (Supplementary Table 2).

3.2 Geochemical results
The results of major ions, nutrients and stable isotopes in different water end
members are shown in Figures 4 and 5. Cl$^{-}$ ranges from 0.6 to 2.1 mg L$^{-1}$ in the
surface water (including riverine inflow water, lake water and downstream water), 0.4
to 2.7 mg L$^{-1}$ in porewater and has a much higher concentration of 5.9 mg L$^{-1}$ in
rainfall water. Na$^{+}$ ranges from 1.6 to 3.4 mg L$^{-1}$ in the surface water, 1.2 to 4.4 mg





$L^{-1}$ in porewater and has a concentration of 4.4 mg $L^{-1}$ in rainfall water. $SO_4^{2-}$ ranges
from 1.2 to 2.3 mg $L^{-1}$ in the surface water, 0.4 to 1.7 mg $L^{-1}$ in porewater and has a
significant low concentration of 0.01 mg $L^{-1}$ in rainfall water. $Ca^{2+}$ ranges from 3.0 to
12.4 mg $L^{-1}$ in lake water, 3.4 to 12.5 mg $L^{-1}$ in porewater and has a significant high
concentration of 20.5 mg $L^{-1}$ in rainfall water. Other concentrations of major ions are
listed in Supplementary Table 2. As shown in Figure 4d and Supplementary Table 2,
$\delta^{18}O$ in the lake water ranges from - 13.06 ‰ to - 12.11 ‰, with an average of -
12.41 ‰ (n = 7), and $\delta^{2}H$ ranges from - 91.83 ‰ to - 87.47 ‰, with an average of -
89.0 ‰ (n = 7). $\delta^{18}O$ in the riverine inflow water ranges from - 13.44 ‰ to - 13.29 ‰,
with an average of – 13.37 ‰ (n = 2), and $\delta^{2}H$ ranges from - 93.25 ‰ to – 91.92 ‰,
with an average of - 92.59 ‰ (n = 2). $\delta^{18}O$ in the downstream water ranges from -
12.51 ‰ to - 12.18 ‰, with an average of - 12.35 ‰ (n = 3), and $\delta^{2}H$ ranges from -
88.96 ‰ to - 87.1 ‰, with an average of - 87.98 ‰ (n = 3). $\delta^{18}O$ in the porewater
ranges from - 12.66 ‰ to - 11.52 ‰, with an average of - 11.97 ‰ (n = 8), and $\delta^{2}H$
ranges from – 91.3 ‰ to -82.87 ‰, with an average of - 85.5 ‰ (n = 8). DIN in the
surface water (including riverine inflow water, lake water and downstream water)
range from 6.6 to 16.9 μM, with an average of 10.3 μM, and DIP from 0.36 to 0.41
μM, with an average of 0.38 μM. The concentrations of DIN for the porewater range
from 0.7 to 358.8 μM, with an average of 92.8 μM, and DIP from 0.18 to 0.44 μM



with an average of 0.31 μM (Figure 5).

**4. Discussion**
4.1 Proglacial hydrologic processes and geochemical implications

Generally, major ion concentrations in the lake water and porewater of Ximen

Co lake are significantly lower than those in main rivers, streams and other tectonic
lakes in the QTP (Wang et al. 2010, Wang et al. 2016b, Yao et al. 2015), and are
similar to those of snow and glaciers (Liu et al. 2011), suggesting that the lake water
is mainly originated from glacier and snow melting. Ion concentrations in the lake and
porewater of Ximen Co lake are much lower than those of rainfall collected in Jiuzhi
town. This suggests that lake water is less influenced by precipitation (Figures 4a-4c).
The concentrations of major ions in the porewater are high compared to the lake water,
indicating weathering affects from the aquifer grains. The ratios of $Ca^{2+}/Na^+$ in the
porewater and groundwater is >1, also suggesting influences of weathering digenesis
of major ions from the seasonal frozen ground at the lake shore aquifer (Wang et al.
2010, Weynell et al. 2016, Yao et al. 2015).

The isotopic compositions of the lake water and porewater are significantly

isotopic depleted, with values close to the compositions of glaciers and surface snow
in the QTP, suggesting the lake is dominantly recharged from snow and glacier

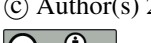



melting (Cui et al. 2014, Wang et al. 2016a, Zongxing et al. 2015). The relation of
$\delta^{18}O$ versus $\delta^2H$ for the lake water is $\delta^2H = 4.25 \times \delta^{18}O - 35.99$, with a slope much
lower than that of the global meteoric water line (GMWL) (Figure 4d), suggesting the
effects of lake surface evaporation. The relation of $\delta^{18}O$ versus $\delta^2H$ for the porewater
is $\delta^2H = 6.93 \times \delta^{18}O - 2.67$, overall on GWML (Figure 4d). Deuterium excesses is
defined as $\Delta D = \delta D - 8 \times \delta^{18}O$ (Dansgaard 1964). The value of $\Delta D$ is dependent on
airmass origins, altitude effect and the kinetic effects during evaporation (Hren et al.
2009). Global meteoric water has a $\Delta D$ of + 10 ‰. In QTP, glacier/snowpack melting
water usually has large positive $\Delta D$, while the precipitations derived from warm and
humid summer monsoon has lower $\Delta D$ (Ren et al. 2017, Ren et al. 2013). In this study,
$\Delta D$ of surface water, lake and porewater ranges from + 37.1 to + 41.2 ‰, closed to
the glacier melting water but much larger than that of the local precipitation of +
29.72 ‰, This indicates the stream and lake water are mainly originated from
glacial/snowpack melting rather than precipitation (Gat 1996, Lerman et al. 1995,
Wang et al. 2016a). The slopes of $\delta^{18}O$ versus $\delta^{18}O$ in lake water and porewater are
4.25 and 6.93, both of which are lower than that of GMWL due to surface evaporation.
Lake water is more intensively influenced by evaporation compared to porewater. The
plots of $\delta^{18}O$ versus $Cl^-$, and $\delta^2H$ versus $Cl^-$ are well clustered for porewater end
member (orange area), lake water end member (blue area), riverine inflow water end



member (yellow area), and precipitation water (Figures 4e and 4f), suggesting stable
$\delta^{18}O$ and $\delta^{2}H$ isotopes and $Cl^-$ can serve as tracers to quantify the hydrologic
partitioning of the lake by setting three endmember models.

The concentrations of DIN and DIP are all within the ranges of other glacial

melting water and proglacial lake water (Hawkings et al. 2016, Hodson 2007, Hodson
et al. 2005, Hudson et al. 2000, Tockner et al. 2002). Briefly, rainfall and upstream
lake water such as YN-4 has the highest DIN concentration, indicating the glacier
melting and precipitation could be important DIN sources in proglacial areas
(Anderson et al. 2017, Dubnick et al. 2017). DIN in porewater is overall higher
compared to the lake water, suggesting the porewater to be DIN effective source; and
DIP concentrations is higher in the lake water compared to porewater, suggesting the
porewater is a DIP sink (Figure 5). The N: P ratios in the lake water and porewater are
averaged to be 27.1 and 320.5, respectively, both much larger than the Redfield Ratio
(N: P = 16:1) in water and organism in most aquatic system and within the range of
other proglacial lakes (Anderson et al. 2017). This also suggests that the lake water
and porewater are under phosphate limited condition. N: P ratio in the rainfall water is
30.4, similar to the lake water. The average N: P ratio of porewater is much higher
than that of lake water, indicting DIN enrichment in the lake shore aquifers (Figure 5).
In pristine groundwater, $NO_3^-$ is the predominated form of N and is highly mobile



within the oxic aquifers, leading to much higher DIN concentrations in the porewater;
DIP has high affinity to the aquifer grains, resulting in much lower DIP concentrations
in the porewater (Lewandowski et al. 2015, Rosenberry et al. 2015, Slomp and Van
Cappellen 2004). Thus, in analogous to surface runoff from glacier/snowpack melting,
LGD can be also regarded as an important source for the proglacial lakes. Because of
very high DIN and N: P ratios in the porewater, a relatively small portion of LGD
delivers considerable nutrients into the glacial lake, shifting the aquatic N: P ratios
and affecting the proglacial aquatic ecosystem (Anderson et al. 2017).

4.2 Estimation of LGD

Figure 6a shows all the sinks and sources of radon with the epilimnion of the lake.

Within $^{222}$Rn transient mass balance model, the dominant $^{222}$Rn loss is atmospheric
degassing/evasion. Generally, $^{222}$Rn degassing rate is the function of the radon-222
concentration gradient at the water-air interface and the parameter of gas piston
velocity $k$, which is finally the function of wind speed and water temperature (Dimova
and Burnett 2011, Gilfedder et al. 2015). To evaluate $^{222}$Rn evasion rate, this study
employs the widely used method proposed by MacIntyre et al. (1995) which is also
detailed described in Supplementary Information. Based on the field data of $^{222}$Rn
concentration in the lake water, wind speed and temperature log, the radon degassing





rate is calculated in a range of 0.8 to 265.2 Bq m$^2$ d$^{-1}$, with an average 42.0 of Bq m$^2$
d$^{-1}$.

In addition to the atmospheric loss and sedimentary diffusion inputs, $^{222}$Rn is also

sinked via radioactive decay, and sourced from decay of parent isotope of $^{226}$Ra. The
decay loss of $^{222}$Rn fluctuates in phase with the distribution of $^{222}$Rn concentration
monitored by RAD 7 AQUA. The equations to estimate benthic fluxes are shown in
supplementary information. The decay loss is calculated to be 26.4 to 223.4 Bq m$^{-2}$ d$^{-1}$,
with an average of 118.0 ± 22.7 Bq m$^{-2}$ d$^{-1}$. $^{226}$Ra concentration is 0.01 Bq m$^{-3}$ for the
lake water. Under secular equilibrium, the $^{226}$Ra decay input can be calculated by
multiplying $^{226}$Ra concentration in the lake water with $\lambda_{222}$ (Corbett et al. 1997, Kluge
et al. 2007, Luo et al. 2016). $^{226}$Ra decay input is calculated to be 0.83 Bq m$^{-2}$ d$^{-1}$,
which is significantly low compared to other $^{222}$Rn sources to the epilimnion.

With the obtained sinks and sources of $^{222}$Rn in the lake, and the constants given in

Table 1, LGD rate can be obtained by dividing the groundwater derived $^{222}$Rn with its
concentration in groundwater endmember. The obtained LGD rate, ranges from -23. 7
mm d$^{-1}$ to 90.0 mm d$^{-1}$, with an average of 10.3 ± 8.2 mm d$^{-1}$ (Figure 7) The LGD rate
range is relatively less than the daily lake water level variations (≈ 50 mm), indicating
that the lake water level variation could be a combined effect of surface runoff and
LGD (Hood et al. 2006). The negative values of LGD rate reflect the return





groundwater flow due to infiltration into the porewater. Normally, the dominant
values are positive, indicating LGD rate is significant compared to water infiltrations
into lakeshore aquifer. The temporal variation of LGD rate could be attributed to the
fluctuations of the hydraulic gradient in the proglacial areas (Hood et al. 2006, Levy
et al. 2015). As indicated by $\Delta$D (mostly > 10) of surface water, the lake and the
upstream water is considered to be mainly recharged from glacial/snowpack melting
rather other precipitations.
To assess the magnitude of uncertainty of $^{222}$Rn transient model, the sensitivity of
estimated LGD to changes in other variables is examined. A sensitivity coefficient $f$ is
proposed to evaluate this uncertainty according to Langston et al. (2013)

$$f = (\Delta F_{LGD} / F_{LGD}) / (\Delta y_i / y_i) \qquad (3)$$

where $\Delta F_{LGD}$ is the amount of change in $F_{LGD}$ from the original value. $\Delta y_i$ is the
amount of change in the other variable of $y_i$ from the original value. Thus, higher $f$
indicates a large uncertainty of final LGD estimate. The uncertainty mainly stems
from $^{222}$Rn measurements in different water endmembers, the atmospheric loss and
water level record. The uncertainties of $^{222}$Rn measurement are about 10 % and 15-20
% in groundwater and lake water endmember, respectively. The uncertainty of
atmospheric loss is derived from uncertainty of $^{222}$Rn in lake water (with an
uncertainty of 15-20 %), temperature (with an uncertainty ≈ 5 %) and wind speed



(with an uncertainty ≈ 5 %). Thus, the final LGD estimate has an uncertainty of

35-40 %.


**4.3 Hydrologic partitioning**

Compared to the groundwater labeled radionuclide of $^{222}$Rn, stable $^{18}$O/$^{2}$H

isotopes are advantageous in the investigation of evaporation processes due to their
fractionations from water to vapor and have been widely used to investigate the
hydrologic cycle of lakes in various environments (Gat 1995, Gibson et al. 1993,
Gonfiantini 1986, Stets et al. 2010). With the field data of stable isotopic composition
and Cl$^{-}$ concentrations in different water end members, groundwater input, surface
water input, lake water outflow and infiltration, and evaporation can be partitioned by
coupling stable isotopic mass balance model with Cl$^{-}$ mass balance model (Figure 6b).

The model, consisting of the budgets of stable isotopes and Cl$^{-}$, and water masses

for the epilimnion, is used to quantify riverine inflow, lake water outflow and
infiltration, and evaporation (Gibson et al. 2016, LaBaugh et al. 1995, LaBaugh et al.
1997). The model is valid under the following assumptions: (1) constant density of
water; (2) no long-term storage change in the reservoir; (3) well-mixed for the
epilimnion (Gibson 2002, Gibson et al. 2016, Gibson and Edwards 2002, LaBaugh et
al. 1997). The above assumptions are reasonably tenable during the short monitoring


period. The model can be fully expressed as
$$F_{in} + F_{LGD} + F_p = F_E + F_{out} \qquad (4)$$

$$F_{in} \times \delta_{in} + F_{LGD} \times \delta_{gw} + F_p \times \delta_p = F_E \times \delta_E + F_{out} \times \delta_L \qquad (5)$$

$$F_{in} \times [Cl^-]_{in} + F_{LGD} \times [Cl^-]_{gw} + F_p \times [Cl^-]_p = F_{out} \times [Cl^-]_L \qquad (6)$$

where $F_{in}$ [mm d$^{-1}$] is the surface water inflow to the lake; $F_{gw}$ [mm d$^{-1}$] is LGD rate.
$F_p$ [mm d$^{-1}$] is the mean daily rainfall rate during the sampling period. $F_E$ [mm d$^{-1}$] is
the lake evaporation. $F_{out}$ [mm d$^{-1}$] is the lake water outflow via runoff and
infiltration into the lake shore aquifer. $\delta_{in}$, $\delta_{gw}$, $\delta_E$ and $\delta_p$ are the isotopic
compositions of surface water inflow, LGD, and evaporative flux, respectively. The
values of $\delta_{in}$, $\delta_{gw}$, and $\delta_p$ are obtained from field data and the composition of $\delta_E$ are
cacluated as shown in supplementary information. $[Cl^-]_{in}$, $[Cl^-]_{gw}$, $[Cl^-]_L$ and
$[Cl^-]_p$ are the chloride concentrations in the inflow water, porewater, lake water and
precipitation, respectively.

The components of the mass balance model can be obtained from the field data of

isotopic composition and Cl$^-$ concentrations in different water endmembers. The
average $^{18}$O composition -13.37 ‰ of riverine inflow water is taken as the value of
the input parameter $\delta_{in}$. $\delta^{18}$O and $\delta^2$H in the groundwater endmember and lake water
end member are calculated to be -12.41 ‰ and -87.18 ‰, respectively. $\delta^{18}$O and $\delta^2$H
in the rainfall are measured to be -5.47 ‰ and -24.98 ‰, respectively. With the





measured values of $\delta_L$, $h$, $\delta_{in}$, and the estimated $\varepsilon$ and $\delta_a$, the isotopic composition
of $\delta_E$ is calculated to be -35.11 ‰, which is in line with the results of alpine and
arctic lakes elsewhere (Gibson 2002, Gibson et al. 2016, Gibson and Edwards 2002).
The values of $[Cl^-]_{in}$, $[Cl^-]_{gw}$, and $[Cl^-]_L$ are calculated to be 0.91 mg L$^{-1}$, 1.48
mg L$^{-1}$ and 1.02 mg L$^{-1}$, respectively. All the parameters used in the model are shown
in Table 2.

According to Equations 4-6, the uncertainties of calculations of $F_{in}$, $F_{out}$ and $E$ are

mainly derived from the uncertainty of $F_{LGD}$ and the compositions of Cl$^-$, $\delta D$ and $\delta^{18}O$
in different water endmembers as suggested in previous studies (Genereux 1998,
Klaus and McDonnell 2013). The compositions of Cl$^-$, $\delta D$ and $\delta^{18}O$ in surface water,
groundwater endmembers have an uncertainty of 5 %. The uncertainty of $\delta_E$ is
reasonably assumed to be $\approx$ 20 % . Thus, considering the uncertainty propagation of
all the above parameters, the uncertainties of $F_{in}$, $F_{out}$ and $E$ would be scaled up to
70-80 % of the final estimates.

4.4 The hydrologic partitioning of the glacial lake

Based on the three endmember model of $^{18}O$ and Cl$^-$, the riverine inflow rate was

calculated to be 135.6 ± 119.0 mm d$^{-1}$, and the lake outflow rate is estimated to be
141.5 ± 132.4 mm d$^{-1}$; the evaporation rate is calculated to be 5.2 ± 4.7 mm d$^{-1}$. The




summary of the hydrologic partitioning of the lake is shown in Figure 8a. Generally,
the proglacial lake is mostly recharged by the riverine inflow from the snowpack or
the glacier melting. The groundwater discharge contributes about only 7.0 % of the
total water input to the lake, indicating groundwater input does not dominate water
input to the proglacial lake. The lake water is mainly lost via surface water outflow
and infiltration to the lake shore aquifers. The evaporation constitutes relatively small
ratio ($\approx$ 3.5 %) of total water losses. The annual evaporation rate was recorded to be
1429.8 mm (equivalent to 3.92 mm d$^{-1}$) in 2015 by the Jiuzhi weather station, lower
than the obtained evaporation in this study. This may be due to much higher
evaporation in August during the monitoring period. The recent review on LGD rate
by Rosenberry et al. (2015) suggests that the median of LGD rate in the literatures is
7.4 mm d$^{-1}$ (0.05 mm d$^{-1}$ to 133 mm d$^{-1}$), which is about 2/3 of LGD rate in this study.
This difference may be due to the hydrogeological setting of the lake shore aquifer.
This aquifer is formed by grey loam, clayey soil and sand (Lehmkuhl 1998, Schlutz
and Lehmkuhl 2009), which is with relatively high permeability. Previous studies
have indicated that groundwater forms a key component of proglacial hydrology
(Levy et al. 2015). However, there have been limited quantitative studies of
groundwater contribution to hydrologic budget of proglacial areas. This study
summarizes the groundwater discharge studies over the glacial forefield areas. Brown


et al. (2006) investigated the headwater streams at the proglacial areas of Taillon
Glacier in French and found that groundwater contributes 6-10 % of the stream water
immediate downwards of the glacier. Using water mass balance model, Hood et al.
(2006) shows that groundwater inflow is substantial in the hydrologic partitioning of
the proglacial Lake O'Hara in front of Opabin Glacier in Canada and comprised of 30
-74 % of the total inflow. Roy and Hayashi (2008) studied the proglacial lakes of
Hungabee lake and Opabin lake at glacier forefield of Opabin Glacier and found that
groundwater component is predominant water sources of the lakes and consisted of
35-39 % of the total water input of the lakes. Langston et al. (2013) further
investigated a tarn immediate in front of Opabin Glacier and indicated the tarn is
predominantly controlled by groundwater inflow/outflow, which consisted of 50-100 %
of total tarn volume. Magnusson et al. (2014) studied the streams in the glacier
forefield of Dammagletscher, Switzerland and revealed that groundwater contributed
only 1-8 % of the total surface runoff. Groundwater contribution in this study is
similar to those obtained the mountainous proglacial areas in Europe, but much lower
than those obtained in the proglacial areas of polar regions. It is concluded that
proglacial lakes/streams in front of mountainous glaciers are mainly recharged by
surface runoff from glacier/snowpack melting. This might be due to well-developed
stream networks and limited deep groundwater flow (Brown et al. 2006, Einarsdottir





et al. 2017, Magnusson et al. 2014). However, proglacial tarns and lakes in the polar
areas are predominantly controlled by groundwater discharge, due to less connectivity
of surface runoff and high shallow and deep groundwater connectivity (Hood et al.
2006, Langston et al. 2013, Roy and Hayashi 2008).
4.5 LGD derived nutrient loadings, nutrient budget and ecological implications

Compared to extensive studies of SGD derived nutrient loadings in the past decade

(Luo and Jiao 2016, Slomp and Van Cappellen 2004), studies of LGD derived nutrient
loadings have received limited attention, even given the fact that groundwater in lake
shore aquifers is usually concentrated in nutrients (Lewandowski et al. 2015,
Rosenberry et al. 2015). Even fewer studies focus on chemical budgets in the
proglacial lakes which are often difficult to access for sampling. Groundwater borne
DIN and DIP across the sediment-water interface in this study are determined with an
equation coupling the advective or LGD-derived, and diffusive solute transport
(Hagerthey and Kerfoot 1998, Lerman et al. 1995)

$$F_j = -nD_j^m \frac{dC_j}{dx} + v_{gw}C_j \qquad (7)$$

where $-nD_j^m \dfrac{dC_j}{dx}$ is the diffusion input and $v_{gw}C_j$ is the LGD derived fluxes, $F_j$
[μM m$^{-2}$ d$^{-1}$] is the mol flux of nutrient species $j$ (representing DIN or DIP). $n$ is the
sediment porosity. $D_j^m$ is the molecular diffusion coefficient of nutrient species $j$,





which is given to be 4.8 x $10^{-5}$ m$^2$ d$^{-1}$ for DIP (Quigley and Robbins 1986), and 8.8 x
$10^{-5}$ m$^2$ d$^{-1}$ for DIN (Li and Gregory 1974), respectively. $C_j$ [μM] is the concentration
of nutrient species $j$. $x$[m] is the sampling depth. $v_{gw}$ is LGD rate estimated by $^{222}$Rn
mass balance model and has a value of 10.3 ± 8.2 mm d$^{-1}$. $\dfrac{dC_j}{dx}$ is the concentration
gradient of nutrient species $j$ across the water-sedimentary interface.
Substituting the constants and the field data of DIN and DIP in to Equation 6, LGD
derived nutrient loadings are calculated to be 954.3 μmol m$^{-2}$ d$^{-1}$ and 3.2 μmol m$^{-2}$ d$^{-1}$
for DIN and DIP, respectively. Riverine inflow brings 1195.0 μmol m$^{-2}$ d$^{-1}$ DIN, 52.9
μmol m$^{-2}$ d$^{-1}$ DIP into the lake. Lake water outflow derived nutrient loss is estimated
to be 1439.9 μmol m$^{-2}$ d$^{-1}$ and 54.7 μmol m$^{-2}$ d$^{-1}$ for DIN and DIP, respectively.
Nutrients in the lake can be also sourced from atmospheric deposit (mostly in form of
precipitation). With the nutrient concentrations in the rain water during the monitoring
period, the wet deposit is calculated to be 76 μmol m$^{-2}$ d$^{-1}$ and 2.5 μmol m$^{-2}$ d$^{-1}$, for
DIN and DIP, respectively. The loadings of DIN to the lakes are mainly from surface
runoff and LGD, which comprised of 42.9 % and 53.7 % of the total DIN loadings..
Groundwater derived DIP input, however, constitutes only 6.3 % of the total DIP
inputs to the lake, indicating groundwater borne DIP is less contributive to the
nutrient budget of the lake compared to DIN. Very recent studies on polar regions
have indicated that the glacier/snowpack water is the main N sources to the proglacial



lakes (Anderson et al. 2013, Dubnick et al. 2017). However, they do not consider the
contribution of groundwater borne N, in spite of the high groundwater connectivity in
the proglacial areas (Roy and Hayashi 2008). This study stresses that groundwater
borne DIN could be comparable to the surface runoff derived DIN.
Based on nutrient results, the lake is considered to be an oligotrophic lake and
under phosphate limited condition. Thus, the primary production (PP) is therefore
considered to be controlled by the DIP loadings. The sum of DIN and DIP inputs
minus the sum of the calculated DIN and DIP outputs leads to surpluses of 785.4
$\mu mol\ m^{-2}\ d^{-1}$ and 3.9 $\mu mol\ m^{-2}\ d^{-1}$ for DIN and DIP, respectively. The surpluses are
expected to be consumed by the phytoplankton and converted into the PP under the
Red Field ratio (C: N: P = 106: 16: 1), leading to a PP of 0.41 mmol C $m^{-2}\ d^{-1}$. The
nutrient budgets for DIN and DIP are summarized in Figures 8b and 8c. The estimated
primary productivity is lower than most temperate eutrophicated and ologotrophic
lakes (Cole et al. 1998, Smith 1979), and comparable to some high latitude or altitude
lakes (Richerson et al. 1986, Sterner 2010).
4.6. Implications, prospective and limitations
Mountainous proglacial lakes are readily developed in glacier forefields of QTP and
other high mountainous glacial such as Europe Alps and Pamir at central Asian
(Heckmann et al. 2016). The proglacial lakes are always trapping system of sediment





and sinks for water and chemical originated from glacier/snowpack melting and
groundwater. In analogous to cosmogenic isotopes such as $^{10}$Be serving as a tool to
quantify the sediment sources, approaches integrating $^{222}$Rn and stable isotopes
provides both qualitatively and quantitatively evaluations of groundwater
contributions and hydrologic partitioning in these remote and untapped lacustrine
systems. Thus, it is expected that the multiple aqueous isotopes is considered to be
effective tools to investigate the LGD and hydrologic partitioning in other proglacial
lakes. This study is mainly limited by the relatively short sampling and monitoring
period. As a special hydrologic regime, the lake shore aquifers of the proglacial lakes
are experiencing frozen-unfrozen transition seasonally, and the dominant recharge of
glacial melting could be fluctuated significantly due to air temperature variation.
Therefore, future groundwater and hydrological studies can be extended to longtime
sampling and monitoring of stable isotopes and $^{222}$Rn in different water endmembers
to reveal the seasonally hydrological and hydrogeological dynamics and their impacts
on local biogeochemical cycles and ecological systems. Special concerns would be
placed on how surface/groundwater interactions and the associated biogeochemical
processes in response to the seasonal frozen ground variations and glacier/snowpack
melting intensity.



## 5. Conclusion


A $^{222}$Rn continuous monitoring is conducted at Ximen Co Lake, a proglacial lake
located at the east QTP. A dynamic $^{222}$Rn mass balance model constrained by radium
mass balance and water level fluctuation is used to quantify temporal distribution of
LGD of the lake. The obtained LGD over the monitoring time ranges from − 23. 7
mm d$^{-1}$ to 80.9 mm d$^{-1}$, with an average of 10.3 ± 8.2 mm d$^{-1}$. Thereafter, a three
endmember model consisting of the budgets of water, stable isotopes and Cl$^-$ is used
to depict the hydrologic partitioning of the lake. Riverine inflow, lake water outflow
via surface runoff, and surface evaporation are estimated to be 135.6 mm d$^{-1}$, 141.5
mm d$^{-1}$ and 5.2 mm d$^{-1}$, respectively. LGD derived nutrient loading is estimated to be
785.4 μmol m$^{-2}$ d$^{-1}$ and 3.2 μmol m$^{-2}$ d$^{-1}$ for DIN and DIP, respectively. Upon
depicting nutrient budget within the lake, the primary productivity is estimated to be
0.41 mol C m$^{-2}$ d$^{-1}$. This study also implicates that LGD constitutes relatively small
portion of the proglacial hydrologic partitioning, however, delivers nearly a half of the
nutrient loadings to the proglacial lake.
This study presents the first attempt to quantify LGD and the associated nutrient
loadings to the proglacial lake of QTP. To our knowledge, there is almost no study on
the groundwater-lake water interaction in the high altitude proglacial lakes in QTP.
This study demonstrates that $^{222}$Rn based approach can be used to investigate the





groundwater dynamics in the high altitude proglacial lakes. The method is
instructional to similar studies in other proglacial lakes in the QTP and elsewhere.

**Acknowledgements**
This study was supported by a grant from the National Natural Science Foundation of
China (NSFC, No.41572208) and the Research Grants Council of Hong Kong Special
Administrative Region, China (HKU17304815). The authors thank Mr. Buming Jiang
for his kind help in the field works during the campaign and Ergang Lian for his help
in stable isotope analysis. The authors thank Jessie Lai for her help in FIA analysis in
School of Biological Sciences, HKU. Supporting data are included as in the files of
supplementary information 2 and 3; Climatological data are purchased through
http://www.weatherdt.com/shop.html; any additional data may be obtained from L.X.
(email: xinluo@hku.hk);

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

Patterns of temporal variation in Lake Titicaca. A high altitude tropical lake. I.
Background, physical and chemical processes, and primary production. Hydrobiologia
897  138(1), 205-220.

Rosenberry, D.O., Lewandowski, J., Meinikmann, K. and Nützmann, G. (2015)
Groundwater-the disregarded component in lake water and nutrient budgets. Part 1:
effects of groundwater on hydrology. Hydrological Processes 29(13), 2895-2921.
Roy, J.W. and Hayashi, M. (2008) Groundwater exchange with two small alpine lakes
in the Canadian Rockies. Hydrological Processes 22(15), 2838-2846.
Schafran, G.C. and Driscoll, C.T. (1993) Flow path-composition relationships for





groundwater entering an acidic lake. Water Resources Research 29(1), 145-154.
Scheidegger, J.M. and Bense, V.F. (2014) Impacts of glacially recharged groundwater
flow systems on talik evolution. Journal of Geophysical Research: Earth Surface
907  119(4), 758-778.

Schlutz, F. and Lehmkuhl, F. (2009) Holocene climatic change and the nomadic
Anthropocene in Eastern Tibet: palynological and geomorphological results from the
Nianbaoyeze Mountains. Quaternary Science Reviews 28(15-16), 1449-1471.
Schmidt, A., Gibson, J.J., Santos, I.R., Schubert, M., Tattrie, K. and Weiss, H. (2010)
The contribution of groundwater discharge to the overall water budget of two typical
Boreal lakes in Alberta/Canada estimated from a radon mass balance. Hydrology and
Earth System Sciences 14(1), 79-89.
Sebok, E., Duque, C., Kazmierczak, J., Engesgaard, P., Nilsson, B., Karan, S. and
Frandsen, M. (2013) High-resolution distributed temperature sensing to detect
seasonal groundwater discharge into Lake Væng, Denmark. Water Resources
Research 49(9), 5355-5368.
Shaw, R.D. and Prepas, E.E. (1990) Groundwater-lake interactions: I. Accuracy of
seepage meter estimates of lake seepage. Journal of Hydrology 119(1–4), 105-120.
Slaymaker, O. (2011) Criteria to distinguish between periglacial, proglacial and
paraglacial environments. Quaestiones Geographicae 30(1), 85-94.
Slomp, C.P. and Van Cappellen, P. (2004) Nutrient inputs to the coastal ocean through
submarine groundwater discharge: controls and potential impact. Journal of
Hydrology 295(1-4), 64-86.
Smerdon, B., Mendoza, C. and Devito, K. (2007) Simulations of fully coupled lake-
groundwater exchange in a subhumid climate with an integrated hydrologic model.
Water Resources Research 43(1).
Smith, V.H. (1979) Nutrient dependence of primary productivity in lakes. Limnology
and Oceanography 24(6), 1051-1064.
Sterner, R.W. (2010) In situ-measured primary production in Lake Superior. Journal of
Great Lakes Research 36(1), 139-149.
Stets, E.G., Winter, T.C., Rosenberry, D.O. and Striegl, R.G. (2010) Quantification of
surface water and groundwater flows to open- and closed-basin lakes in a
headwaters watershed using a descriptive oxygen stable isotope model. Water
Resources Research 46(3).
Tockner, K., Malard, F., Uehlinger, U. and Ward, J. (2002) Nutrients and organic matter
in a glacial river-floodplain system (Val Roseg, Switzerland). Limnology and
Oceanography 47(1), 266-277.
Valiela, I., Teal, J.M., Volkmann, S., Shafer, D. and Carpenter, E.J. (1978) Nutrient and
Particulate Fluxes in a Salt-Marsh Ecosystem - Tidal Exchanges and Inputs by





Precipitation and Groundwater. Limnology and Oceanography 23(4), 798-812.
Wang, C., Dong, Z., Qin, X., Zhang, J., Du, W. and Wu, J. (2016a) Glacier meltwater
runoff process analysis using $\delta$D and $\delta^{18}$O isotope and chemistry at the remote
Laohugou glacier basin in western Qilian Mountains, China. Journal of Geographical
Sciences 26(6), 722-734.
Wang, J., Zhu, L., Wang, Y., Ju, J., Xie, M. and Daut, G. (2010) Comparisons between
the chemical compositions of lake water, inflowing river water, and lake sediment in
Nam Co, central Tibetan Plateau, China and their controlling mechanisms. Journal of
Great Lakes Research 36(4), 587-595.
Wang, R., Liu, Z., Jiang, L., Yao, Z., Wang, J. and Ju, J. (2016b) Comparison of surface
water chemistry and weathering effects of two lake basins in the Changtang Nature
Reserve, China. Journal of Environmental Sciences 41, 183-194.
Wang, S. (1997) Frozen ground and environment in the Zoige Plateau and its
surrounding mountains (In Chinese with English abstract). Journal of Glaciology and
Geocryology 19(1), 39-46.
Warner, N.R., Christie, C.A., Jackson, R.B. and Vengosh, A. (2013) Impacts of shale gas
wastewater disposal on water quality in western Pennsylvania. Environmental
Science & Technology 47(20), 11849-11857.
Weynell, M., Wiechert, U. and Zhang, C. (2016) Chemical and isotopic (O, H, C)
composition of surface waters in the catchment of Lake Donggi Cona (NW China) and
implications for paleoenvironmental reconstructions. Chemical Geology 435, 92-107.
White, D., Lapworth, D.J., Stuart, M.E. and Williams, P.J. (2016) Hydrochemical
profiles in urban groundwater systems: New insights into contaminant sources and
pathways in the subsurface from legacy and emerging contaminants. Science of the
Total Environment 562, 962-973.
Wilson, J. and Rocha, C. (2016) A combined remote sensing and multi-tracer
approach for localising and assessing groundwater-lake interactions. International
Journal of Applied Earth Observation and Geoinformation 44, 195-204.
Winter, T.C. (1999) Relation of streams, lakes, and wetlands to groundwater flow
systems. Hydrogeology Journal 7(1), 28-45.
Wischnewski, J., Herzschuh, U., Rühland, K.M., Bräuning, A., Mischke, S., Smol, J.P.
and Wang, L. (2014) Recent ecological responses to climate variability and human
impacts in the Nianbaoyeze Mountains (eastern Tibetan Plateau) inferred from
pollen, diatom and tree-ring data. Journal of Paleolimnology 51(2), 287-302.
Xin, W., Yongjian, D., Shiyin, L., Lianghong, J., Kunpeng, W., Zongli, J. and Wanqin, G.
(2013) Changes of glacial lakes and implications in Tian Shan, central Asia, based on
remote sensing data from 1990 to 2010. Environmental Research Letters 8(4),
979     044052.





Yao, T., Thompson, L., Yang, W., Yu, W., Gao, Y., Guo, X., Yang, X., Duan, K., Zhao, H.,
Xu, B., Pu, J., Lu, A., Xiang, Y., Kattel, D.B. and Joswiak, D. (2012) Different glacier
status with atmospheric circulations in Tibetan Plateau and surroundings. Nature
Clim. Change 2(9), 663-667.
Yao, Z., Wang, R., Liu, Z., Wu, S. and Jiang, L. (2015) Spatial-temporal patterns of
major ion chemistry and its controlling factors in the Manasarovar Basin, Tibet.
Journal of Geographical Sciences 25(6), 687-700.
Yuan, H., Liu, E., Shen, J., Zhou, H., Geng, Q. and An, S. (2014) Characteristics and
origins of heavy metals in sediments from Ximen Co Lake during summer monsoon
season, a deep lake on the eastern Tibetan Plateau. Journal of Geochemical
Exploration 136, 76-83.
Zhang, C. and Mischke, S. (2009) A Lateglacial and Holocene lake record from the
Nianbaoyeze Mountains and inferences of lake, glacier and climate evolution on the
eastern Tibetan Plateau. Quaternary Science Reviews 28(19), 1970-1983.
Zlotnik, V.A., Olaguera, F. and Ong, J.B. (2009) An approach to assessment of flow
regimes of groundwater-dominated lakes in arid environments. Journal of Hydrology
996   371(1), 22-30.
Zlotnik, V.A., Robinson, N.I. and Simmons, C.T. (2010) Salinity dynamics of discharge
lakes in dune environments: Conceptual model. Water Resources Research 46(11).
Zongxing, L., Qi, F., Wei, L., Tingting, W., Xiaoyan, G., Zongjie, L., Yan, G., Yanhui, P.,
Rui, G. and Bing, J. (2015) The stable isotope evolution in Shiyi glacier system during
the ablation period in the north of Tibetan Plateau, China. Quaternary International
1002  380, 262-271.




**Figure captions**

**Figure 1** The geological and topographic map of the Yellow River Source Region,
Nianbaoyeze glacial mountains (**a**), and the sampling settings of the Ximen Co Lake
(**b**), with the bathymetry map of the lake (**d**). (**c**) Photograph of the Ximen Co Lake
and the surrounding geomorphic settings looking northeast direction on 22 Aug 2015,
showing the late-laying snowpack in the U-shaped valleys of the north part of
Nianbaoyeze MT.





**Figure 2** The climatological parameters (wind speed, air temperature, and precipitation) in the Aug, 2015 recorded from Jiuzhi weather station.

**Figure 3** The temporal distributions of $^{222}$Rn (**a**), water temperature (**b**), water level fluctuation recorded by the divers (**c**), and hourly wind speed and air temperature recorded in Jiuzhi weather station (**d**).

**Figure 4** The cross plots of Cl$^-$ versus Na$^+$ (**a**), SO$_4^{2-}$ versus Cl$^-$ (**b**), Ca$^{2+}$ versus Cl$^-$ (**c**); The relations of $^2$H versus $^{18}$O (**d**), Cl$^-$ versus $^2$H (**e**), and Cl$^-$ versus $^{18}$O (**f**).

**Figure 5** Cross plots of $^{222}$Rn versus DIN (**a**) and DIP (**b**).

**Figure 6** The conceptual model of $^{222}$Rn transient model (**a**), and three endmember model (**b**).

**Figure 7** The results of the final LGD derived from $^{222}$Rn transient model.

**Figure 8** The hydrologic partition of the proglacial lake of Ximen CO (**a**), and the budgets of DIN (**b**) and DIP (**c**).



Figure 1

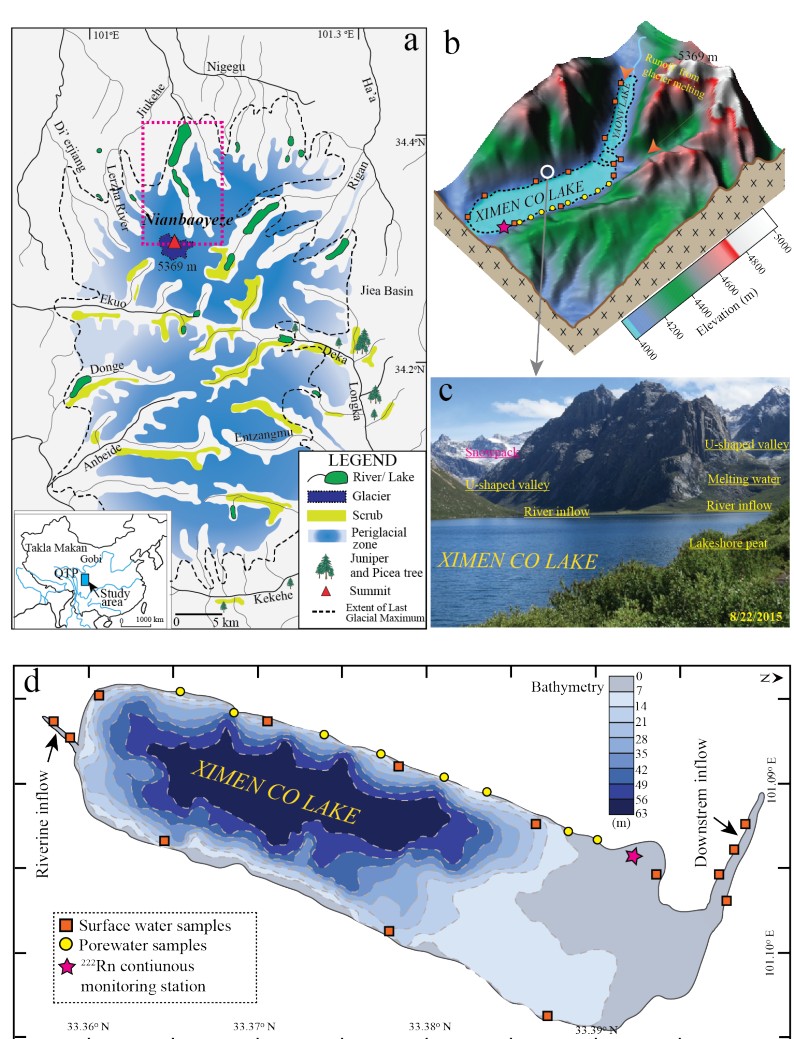










Figure 2

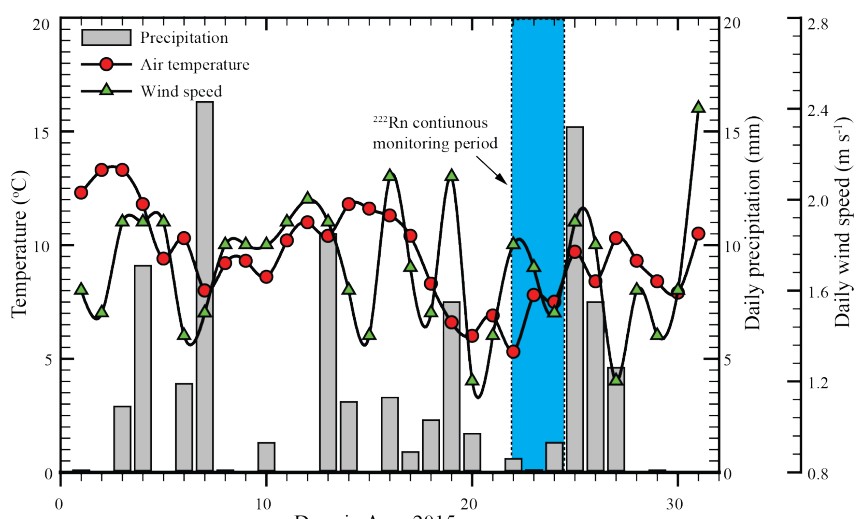

















Figure 3


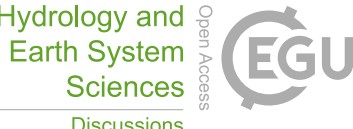



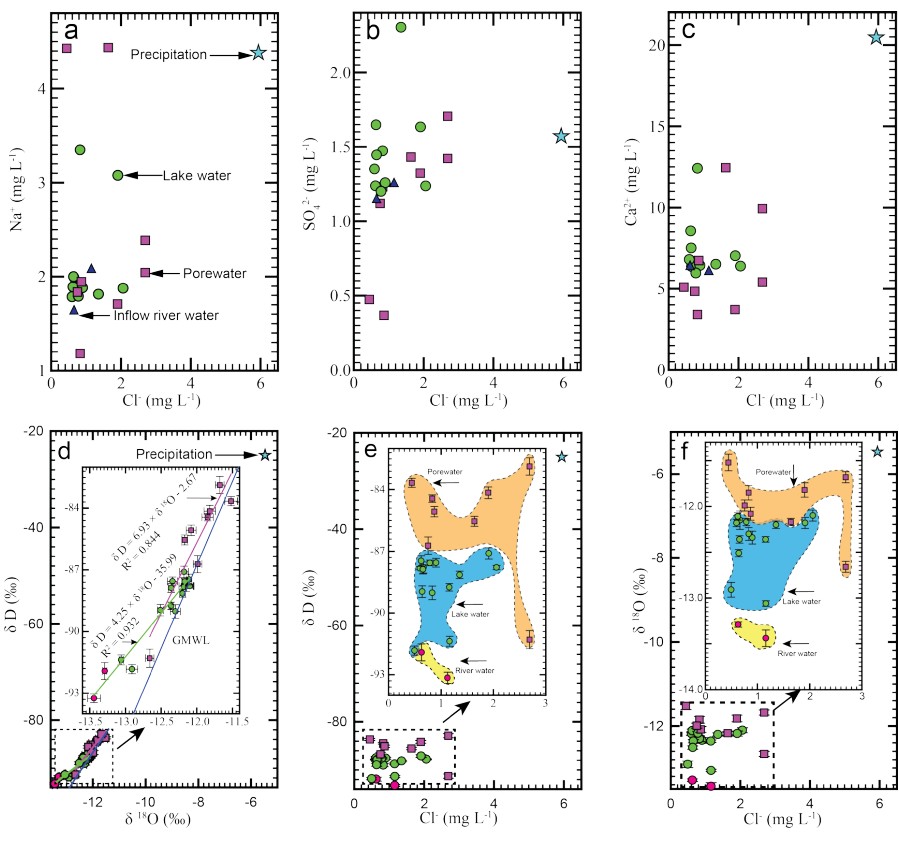












Figure 6

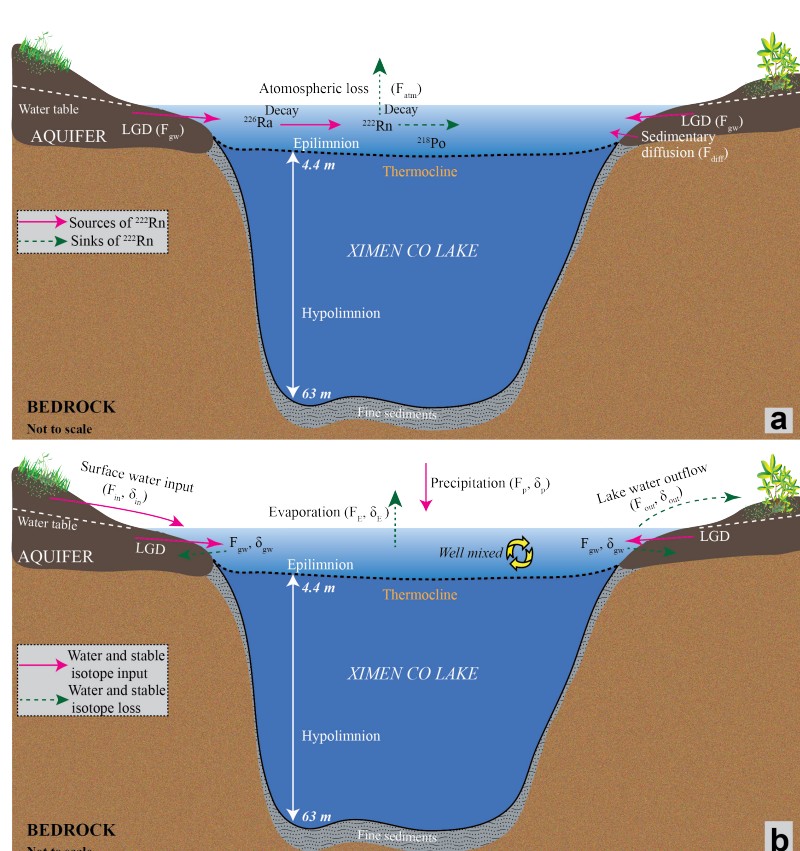











Figure 7

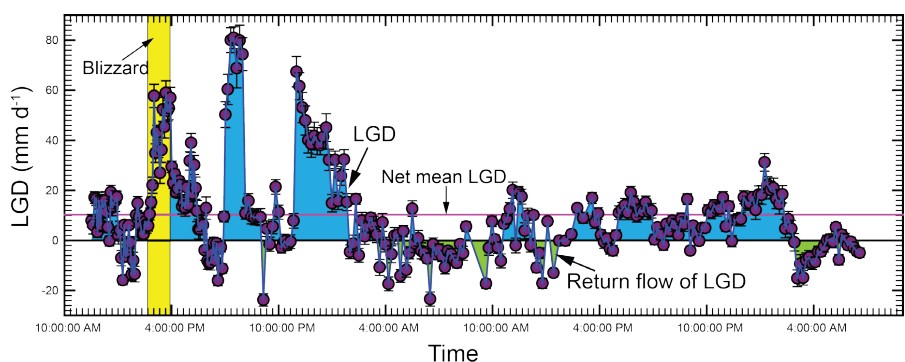

















Figure 8

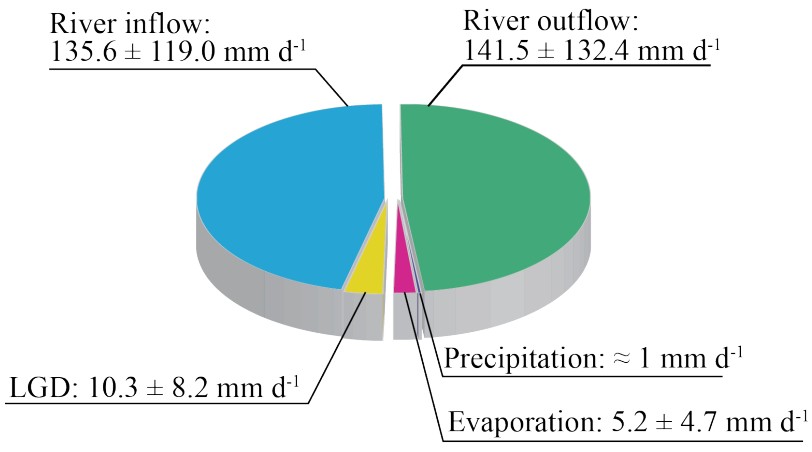

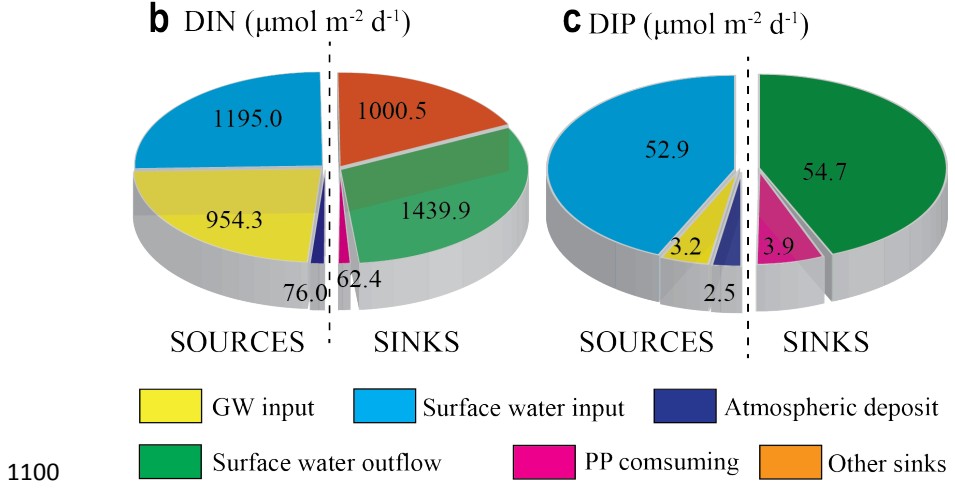




**Table 1** Parameters used to establish the mass balance model of $^{222}$Rn in Ximen Co Lake.

| Parameters | Units or values | Estimated Uncertainty (%) | Evaluation |
|---|---|---|---|
| Wind speed ($\omega_{10}$) | 0.2 - 5.4 m s$^{-1}$ | n.a | From Jiuzhi weather stations |
| Water-air temperature | 11.2 - 15.6 °C | n.a | Recorded with probe in the chamber; sensitive to temperature results |
| Molecular diffusion of $^{222}$Rn in water ($D_m$) | 9.2×10$^{-6}$ - 1.0×10$^{-5}$ cm$^2$ s$^{-1}$ | n.a | 1.16×10$^{-6}$ at 20 °C; adjustable for temperature |
| Molecular diffusion of $^{222}$Rn in sediments ($D_s$) | 2.2×10$^{-6}$ - 2.5×10$^{-5}$ cm$^2$ s$^{-1}$ | n.a | Adjusted for temperature, sediment porosity |
| Dynamic viscosity ($\mu$) | 1.1×10$^{-3}$ - 1.3×10$^{-3}$ cm$^2$ s$^{-1}$ | n.a | Calculated based on water temperature, density and salinity |
| Schmidt number ($S_c$) | 1078.6 - 1371.6 [ - ] | n.a | Calculated as the ratio of $\upsilon$ to $D_m$ |
| Water depth ($H$) | 4.4 m | n.a | Epilimnion depth of Ximen Co Lake |
| Decay constant $^{222}$Rn ($\lambda_{222}$) | 0.186 d$^{-1}$ | n.a | Constant |
| Groundwater endmember $^{222}$Rn ($C_{gw}$) | 11200 ± 1200 Bq m$^{-3}$ | 8 sites dependant for | Measured; final result for water flux inversely proportional to $^{222}$Rn |
| Lake water endmember $^{222}$Rn ($C_l$) | 21.6 - 418.8 Bq m$^{-3}$ | 15-25% | Measured with RAD 7 AQUA |
| Atmospheric $^{222}$Rn ($C_a$) | 1.5 ± 1.0 Bq m$^{-3}$ | 20-25% | Measured or assumed value, model not sensitive to radon in air variation |
| K$_{air/water}$ | 0.29 - 0.33 [ - ] | n.a | Calculated based on temperature in the chamber and salinity in lake water |
| Porosity $n$ | 0.31 | n.a | Assumed based on literatures |
| Tortuosity $\theta$ | 2.05 | n.a | Calculated based on porosity |
| Piston velocity ($\kappa$) | 0.004 - 1.11 m d$^{-1}$ | 20-25% | Calculated from Equation 3 in supplementary information |
| $^{226}$Ra concentration in lake waters ($C_{226Ra}$) | 0.01 Bq m$^{-3}$ | ≈10% | Measured with RAD7 |
| Diffusive flux of $^{222}$Rn ($F_{diff}$) | 0.68 - 213.5 Bq m$^{-2}$ d$^{-1}$ | n.a | Calculated from Equation 9 in supplementary information |
| Atmospheric flux of $^{222}$Rn ($F_{atm}$) | 0.7 - 213.5 Bq m$^{-2}$ d$^{-1}$ | n.a | Calculated from Equation 1 in supplementary information |
| Groundwater flux of $^{222}$Rn ($F_{gw}$) | 14.7 - 349.8 Bq m$^{-2}$ d$^{-1}$ | n.a | Calculated from Equation 1 |
| Inventory of $^{222}$Rn ($I$) | Bq m$^{-2}$ | n.a | Measured with RAD7 AQUA |
| Groundwater discharge ($Q_{gw}$) | 10.3 ± 8.2 (3.5-38.6) mm d$^{-1}$ | n.a | Calculated from Equation 1 |


**Table 2** Input parameters for the three endmember model of Ximen Co Lake

| Input parameter | Description | Values (using $^{18}$O as a tracer) | Parametric sources |
|---|---|---|---|
| $h$ | Relatively humidity | 0.63 | Measured by the humidity meter |
| $T$ (°C) | Water temperature | 15.66 | Monitored with divers |
| $\delta_{surface}$ ($^{18}$O) ‰ | Surface water isotopic compositions | -12.45 | Average value of surface inflow samples |
| $\delta_{gw}$ ($^{18}$O) ‰ | Groundwater isotopic compositions | -11.97 | Average value of porewater samples |
| $\delta_L$ ($^{18}$O) ‰ | Lake water isotopic compositions | -12.54 | Average value of Ximen Co Lake water samples |
| $F_{gw}$ (mm/d) | LGD rates | 14.18 | Calculated based on $^{222}$Rn mass balance model |
| $\varepsilon^*$ ($^{18}$O) ‰ | Effective equilibrium isotopic enrichment factor | 10.12 | Equations 13-14 in supplementary information |
| $C_k$ ($^{18}$O) ‰ | Kinetic constant for $^{18}$O | 14.2 | Constants based on evaporating experiment |
| $\varepsilon_k$ ($^{18}$O) ‰ | Kinetic enrichment factor | 5.2 | From Equation 15 in supplementary information |
| $\varepsilon$ ($^{18}$O) ‰ | Total isotopic enrichment factor | 15.33 | The sum of $\varepsilon^*$ and $C_k$ |
| $\alpha^*$ ($^{18}$O) ‰ | Effective isotopic equilibrium factor | 1.01 | $\alpha^*=1+\varepsilon^*$ |
| $\delta_a$ ($^{18}$O) ‰ | Isotopic composition of ambient air | -23.12 | Estimated with $\delta_{in}$ and $\delta_a$ |
| $\delta_{in}$ ($^{18}$O) ‰ | Isotopic composition of surface inflow water | -13.41 | Average value of surface inflow water |
| $\delta_E$ ($^{18}$O) ‰ | Isotopic compositions of evaporating vapor | -35.1 | From Equation 12 in supplementary information |
| $[Cl]_{in}$ (mgL$^{-1}$) | Chloride concentrations in surface inflow water | 0.91 | Filed data |
| $[Cl]_L$ (mgL$^{-1}$) | Chloride concentrations in lake water | 1.02 | Filed data |
| $[Cl]_{gw}$ (mgL$^{-1}$) | Chloride concentrations in groundwater | 1.48 | Filed data |