# Peer review of "Revised MS submitted to HESS"

_Hydrology and Earth System Sciences, 2018_

## Referee Comment (RC1) · Anonymous Referee #1 · 12 Mar 2018

General comments: This paper is focusing on the evaluation of LGD and its related nutrient budgets and hydrologic partitioning in proglacial lake of QTP. The work is great and the paper is overall well organized. Anyway, I have the following comments for the authors to consider.

Specific comments 1. Authors should address more about why it's important to study proglacial lake, especially the ones in QTP, in the introduction part. 2. The primary productivity is calculated based on the dissolved inorganic nutrient budgets. Authors should be careful to do so. Did the authors consider the transformation between

dissolved inorganic and particulate inorganic forms? Redfield ratio usually works in oceanic aquatic system. In lakes, the ratio is fairly variable. 3. Radon in air is important term to do balance calculation. Was the Rn in air measured? I did not see the information or data about this term in the manuscript or the SI. 4. Line 53-60, these two sentences are both started with the locations. Please revise them. 5. Line 208, how often were the Ra-226 samples collected? Or just one sample, and you assume Ra-226 is constant? 6. Line 279, how long is ïĄĎt? 7. Line 327, figure 5 is not attached.

Technical corrections 1. Line 144, 0.7 or 7? 2. Line 195, the unit should be L min-1. 3. Line 230, change "recently" to "recent". 4. Line 280, should be Equation (2). 5. Line 383, two 18O?

–––––––––––––––––––––––––––––

---

## Referee Comment (RC2) · Anonymous Referee #2 · 25 Mar 2018

This is an interesting and generally well-written paper that makes a good contribution to understanding the groundwater surface water interactions and estimating the lacustrine groundwater discharge in mountainous proglacial lakes in the QTP. The abstract is correctly informative with some remarks (see below). The introduction and the site description take into account previous papers in exhaustive way. The methodological approach for data analysis is modern without particular novelties. High-resolution 222Rn activities, water level, both in water temperature and ,wind speed together with stable isotopic data are quite impressive.Many studies have done to explain the high

222Rn concentrations in groundwater near the coast, with the intent of defining a hydrogeological investigation method, which can also be used for coastal aquifers by means of 222Rn and its ancestor 226Ra. However,it is rare to use this method on the QTP. It is suitable for publication in HESS following moderate revision as outlined below. I think that the paper requires a) A rethink abut what material is strictly necessary in Section 4 and/or better guidance to how the information addresses the main points of the paper. This is probably the major concern. b) More consideration as to how this study can inform others elsewhere in the world. I hope that the comments are useful to the authors in revising this study.

Abstract Line 36: DIN and DIP should be written in full name, rather than the abbreviation. Lines 38-39: Not clear what you mean by this.

Introduction Lines 56-57: the citations should be shown in the chronological order. Check them in the whole text.

2.1 Sit descriptions Line 149: 4030 m asl, use the full term when it first occurs. Line 161: 4150 m changes to 4150 m.a.s.l. Line 173: Does the 'pore water'refer to groundwater? What types are the sampling wells? What depths do the wells pump from?

2.2 Sampling and field analysis & 2.3 Chemical analysis (should be "Chemical and isotopic analysis") Quote the precision for all of the parameters and lower detection limits where important.

Lines 192-210: This is a standard technique and the description of it could be shortened.

Lines 337-339:"$\delta$18O in the lake water ranges from -13.06‰ to -12.11‰ with an average of -12.41‰ (n=7), and $\delta$2H ranges from -91.83‰ to -87.47‰ with an average of -89.0‰ (n=7)." should be changed to "$\delta$18O in the lake water ranges from -13.1‰ to -12.1‰ with an average of -12.4‰ (n=7), and $\delta$2H ranges from -92‰ to -87‰ with an average of -89‰ (n=7)." Keep $\delta$18O values in one after the decimal point and $\delta$2H in

single digits in the whole paper.

4.Discussion Do the adjoining lacustrine aquifers receive ('recharge') to sustain the inferred rate of groundwater discharge? And is the inferred width of the zone of lacustrine groundwater discharge compatible with the physics of the groundwater flow system and hydrological cycle? Did you consider the lag time between recharge and chemical changes in the lacustrine aquifers? Please consider the relationship between Fig 5 and Fig 6 to give a relevant illustration on chemical components and isotopic data.

Fig. 6 The conceptual model of 222Rn transient model looks well. But the associated illustration in the text is not convincing on the flow pathways for the 222Rn sources. Clearly some components of the conceptual understanding are not supported by the data. The manuscript would also benefit greatly from a more thorough literature review, which in-turn will help establish the objectives of the work.

My main concern with the paper is with the 222Rn analysis that I don't think is well enough explained to be convincing. Doing a more thorough job on this will add material.

Conclusions This section just summarizes the main findings of the project. In the introduction you make some general statements about the need to understand processes in these impacted lacustrine aquifers in general. In this section explain in more detail how your project helps us to understand processes in these environments more broadly; the paper will have more impact if researchers from elsewhere in the world can see relevance to their studies and a paper in a major international journal such as HESS needs to have broad appeal.

---

## Author Comment (AC1) · 7 May 2018

Reviewer #1 General comments: This paper is focusing on the evaluation of LGD and its related nutrient budgets and hydrologic partitioning in proglacial lake of QTP. The work is great and the paper is overall well organized. Anyway, I have the following comments for the authors to consider.

Reply to General comments: ïČŸ We appreciate the overall positive comments for this study.

[Figure]

Specific comments 1. Authors should address more about why it's important to study proglacial lake, especially the ones in QTP, in the introduction part. ïČŸ Well taken. We have added more description of the proglacial lakes, and lake dynamics in QTP under the influence of climate change and global warming. Some latest lake studies in the QTP were also reviewed (lines 72-93).

2. The primary productivity is calculated based on the dissolved inorganic nutrient budgets. Authors should be careful to do so. Did the authors consider the transformation between dissolved inorganic and particulate inorganic forms? Redfield ratio usually works in oceanic aquatic system. In lakes, the ratio is fairly variable. ïČŸ Good question, indeed, we noticed that the DIN and DIP relation cannot be used to quantify the primary production in the fresh lacustrine systems due to the high variability of Redfield ratios and the possibility of transformation between DIN/DIP. Based on the measurement results, DIN: DIP ratios in the lake water and groundwater are all much large than 16:1, indicating the phosphate limited conditions. ïČŸ As indicated by previous studies of glacial melting water bodies in the QTP, Arctic and Antarctic, the dominant form of dissolve phosphate is DIP, while the DOP contributes less than 10 % of the dissolved phosphate (Hawkings et al. 2016, Hodson 2007, Hodson et al. 2005, Liu et al. 2011, Mitamura et al. 2003). In the freshwater system, particular? phosphate is highly bounded. Therefore, it is reasonable to assume that primary production in the lake water is limited by DIP, and to assign glacial melting water as phosphate limited condition in this study. ïČŸ The primary producers in the lake system consume the nutrient production under variant N: P ratios as indicated by previous studies (Downing and McCauley 1992). To avoid the ambiguous statements and conclusions, we discussed this term as the biological uptake/transformation of nutrients, and removed the ambiguous statements on primary production throughout the MS (lines 707-751).

3. Radon in air is important term to do balance calculation. Was the Rn in air measured? I did not see the information or data about this term in the manuscript or the SI. ïČŸ Yes, we placed RAD7 for air radon-222 measurement at the lake shore for about

4 hours, and the mean activity of the lake area is $1.51 \pm 0.97$ Bq m-3. We added the ambient air radon-222 activities in Table 1 and some descriptions in the methodology part (lines 249-252).

4. Line 53-60, these two sentences are both started with the locations. Please revise them. ïČŸ Well taken, these sentences were revised as suggested (lines 67-68).

5. Line 208, how often were the Ra-226 samples collected? Or just one sample, and you assume Ra-226 is constant? 6. Line 279, how long is ïAËŻDĚĞ t? 7. Line 327, figure 5 is not attached. ïČŸ Due to the bad weather condition and large sampling volume, we could only obtain one 226Ra sample. However, radium in the lacustrine system is significantly lower compared to 222Rn, therefore, the decay input of 222Rn from 226Ra is minimal and negligible. Thus, the spatial heterogeneity of radon-222 will mount minimal effects on the final 222Rn mass balance models. ïČŸ The time step is set to be 5 min, in consistent with the 222Rn record interval. More statements? were added in the revised MS (lines 333-334). ïČŸ Figure 5 was attached in the revised MS.

Technical corrections 1. Line 144, 0.7 or 7? ïČŸ Well taken, this is 0.7 m s-1, and change was made in the revision MS (line 187).

2. Line 195, the unit should be L min-1. ïČŸ Well taken, unit was change to L min-1 (line 238).

3. Line 230, change "recently" to "recent". ïČŸ Well taken, change was made (line 283).

4. Line 280, should be Equation (2). ïČŸ Well taken, change was made (line 335).

5. Line 383, two 18O? ïČŸ Typo and change was made (line 439).

References Downing, J.A. and McCauley, E. (1992) The nitrogen: phosphorus relationship in lakes. Limnology and Oceanography 37(5), 936-945. Hawkings, J., Wadham, J., Tranter, M., Telling, J., Bagshaw, E., Beaton, A., Simmons, S.L., Chandler, D., Tedstone, A. and Nienow, P. (2016) The Greenland Ice Sheet as a

hot spot of phosphorus weathering and export in the Arctic. Global Biogeochemical Cycles. Hodson, A. (2007) Phosphorus in glacial meltwaters. Glacier Science and Environmental Change, 81-82. Hodson, A., Mumford, P., Kohler, J. and Wynn, P.M. (2005) The High Arctic glacial ecosystem: new insights from nutrient budgets. Biogeochemistry 72(2), 233-256. Liu, Y., Yao, T., Jiao, N., Tian, L., Hu, A., Yu, W. and Li, S. (2011) Microbial diversity in the snow, a moraine lake and a stream in Himalayan glacier. Extremophiles 15(3), 411-421. Mitamura, O., Seike, Y., Kondo, K., Goto, N., Anbutsu, K., Akatsuka, T., Kihira, M., Tsering, T.Q. and Nishimura, M. (2003) First investigation of ultraoligotrophic alpine Lake Puma Yumco in the pre-Himalayas, China. Limnology 4(3), 167-175.

Please also note the supplement to this comment:
https://www.hydrol-earth-syst-sci-discuss.net/hess-2018-26/hess-2018-26-AC1-supplement.pdf
* * *

---

## Author Comment (AC2) · 7 May 2018

Reviewer #2 General comments: This is an interesting and generally well-written paper that makes a good contribution to understanding the groundwater surface water interactions and estimating the lacustrine groundwater discharge in mountainous proglacial lakes in the QTP. The abstract is correctly informative with some remarks (see below). The introduction and the site description take into account previous papers in exhaustive way. The methodological approach for data analysis is modern without particular novelties. High-resolution 222Rn activities, water level, both in water temperature and,

wind speed together with stable isotopic data are quite impressive. Many studies have done to explain the high single digits in the whole paper.  We appreciated the overall positive comments.

4. Discussion Do the adjoining lacustrine aquifers receive ('recharge') to sustain the inferred rate of groundwater discharge? And is the inferred width of the zone of lacustrine groundwater discharge compatible with the physics of the groundwater flow system and hydrological cycle?  The regional precipitation recorded by the Jiuzhi station is to be around 90 mm d-1 during Aug, 2015. When deploying an empirical infiltration coefficient of 0.2 for the lake basin, the aquifer recharge rates are yielded up to 18 mm d-1, which is sufficient to maintain the water balance within the lacustrine aquifers. Moreover, as indicated by previous studies in an interior lake of the QTP, Nam Co, a lake located at the area with relatively high evaporation and lower precipitation, its LGD is estimated to be 5-8 mm d-1 and is comparable to the results of this study. This also indirectly implicate that the LGD in this study is tenable and can be balanced by the recharge of the lacustrine aquifers, as Ximen Co basin is influenced by rather larger precipitation and lower evaporation compared to Nam Co.  The inferred width of the zones of lacustrine groundwater discharge is also regarded as the seepage face. Previous studies have indicated that the groundwater seepage areas are mostly located along the transect within 10-20 m across the lakeshore (Luo et al. 2016, Luo et al. 2017, Rosenberry et al. 2015, Schafran and Driscoll 1993). While the deep groundwater system is rather constrained by the Precambrian bedrocks (Einarsdottir et al. 2017), and the LGD occurrence is considered to be constrained within the seepage faces along the lakeshores, and within the bathymetry of epilimnion.

Did you consider the lag time between recharge and chemical changes in the lacustrine aquifers?  The lag time between the recharge and the chemical changes in the lacustrine aquifers is not considered in this study for the following reasons: (1) For 222Rn, the equilibrium state is assigned as 222Rn will reach equilibrium states within short distance (sever centimeters) and elapsed time after the infiltration (Ku et al. 1992,

Porcelli 2008). (2) Stable isotopes generally behave rather conservatively after entering the aquifer and there is negligible fractionation during the transport in the aquifer between the recharge and discharge. (3) The groundwater sampling locations were located at the immediate zones of the lake shore, and therefore, the dynamics the flow length and recharge lag time is minimal and negligible for the reactive solutes of DIN and DIP, similar as suggested in many previous studies (Dimova and Burnett 2011, Dimova et al. 2013, Kluge et al. 2012, Luo et al. 2016).

Please consider the relationship between Fig 5 and Fig 6 to give a relevant illustration on chemical components and isotopic data.  We are sorry that we forgot to attach Figure 5 in the previous version. This figure was used to give a relevant illustration on chemical components and isotopic data.

Fig. 6 The conceptual model of 222Rn transient model looks well. But the associated illustration in the text is not convincing on the flow pathways for the 222Rn sources. Clearly some components of the conceptual understanding are not supported by the data. The manuscript would also benefit greatly from a more thorough literature review, which in-turn will help establish the objectives of the work. My main concern with the paper is with the 222Rn analysis that I don't think is well enough explained to be convincing. Doing a more thorough job on this will add material.  If we understood properly, this comment has two points: reliability of some components and literature review. The reviewer did not specify which components that were no supported by the data. We guess they could be lake evaporation and riverine inflow. To take account this comment, we have reviewed more relevant literatures and added more discussion on lake evaporation and riverine inflow (lines 586 to 664).

Conclusions This section just summarizes the main findings of the project. In the introduction you make some general statements about the need to understand processes in these impacted lacustrine aquifers in general. In this section explain in more detail how your project helps us to understand processes in these environments more broadly; the paper will have more impact if researchers from elsewhere in the world

can see relevance to their studies and a paper in a major international journal such as HESS needs to have broad appeal  To stress the research significance of this study, we have added more discussions to explain how the results of this study facilitate the understanding the environments more broadly (lines 767-773). We hope the updated MS can meet the board research interest of HESS.

References Dimova, N.T. and Burnett, W.C. (2011) Evaluation of groundwater discharge into small lakes based on the temporal distribution of radon-222. Limnol. Oceanogr 56(2), 486-494. Dimova, N.T., Burnett, W.C., Chanton, J.P. and Corbett, J.E. (2013) Application of radon-222 to investigate groundwater discharge into small shallow lakes. Journal of Hydrology. Einarsdottir, K., Wallin, M.B. and Sobek, S. (2017) High terrestrial carbon load via groundwater to a boreal lake dominated by surface water inflow. Journal of Geophysical Research: Biogeosciences, n/a-n/a. Kluge, T., von Rohden, C., Sonntag, P., Lorenz, S., Wieser, M., Aeschbach-Hertig, W. and Ilmberger, J. (2012) Localising and quantifying groundwater inflow into lakes using high-precision< sup> 222 Rn profiles. Journal of Hydrology. Ku, T.-L., Luo, S., Leslie, B. and Hammond, D. (1992) Uranium-series disequilibrium: applications to earth, marine, and environmental sciences. 2. ed. Luo, X., Jiao, J.J., Wang, X.-s. and Liu, K. (2016) Temporal 222 Rn distributions to reveal groundwater discharge into desert lakes: implication of water balance in the Badain Jaran Desert, China. Journal of Hydrology 534, 87-103. Luo, X., Jiao, J.J., Wang, X.-s., Liu, K., Lian, E. and Yang, S. (2017) Groundwater discharge and hydrologic partition of the lakes in desert environment: Insights from stable 18 O/2 H and radium isotopes. Journal of Hydrology 546, 189-203. Porcelli, D. (2008) Investigating groundwater processes using U-and Th-series nuclides. Radioactivity in the Environment 13, 105-153. Rosenberry, D.O., Lewandowski, J., Meinikmann, K. and Nützmann, G. (2015) Groundwater‐the disregarded component in lake water and nutrient budgets. Part 1: effects of groundwater on hydrology. Hydrological Processes 29(13), 2895-2921. Schafran, G.C. and Driscoll, C.T. (1993) Flow path‐composition relationships for groundwater entering an acidic lake. Water Resources Research 29(1), 145-154.

Please also note the supplement to this comment:
https://www.hydrol-earth-syst-sci-discuss.net/hess-2018-26/hess-2018-26-AC2-supplement.pdf

---

## Author Comment (AC3) · 7 May 2018

[revised manuscript text omitted]

Figure 2

[Figure]

Figure 3

[Figure]

Figure 4

[Figure]

Figure 5

[Figure]

Figure 6

.
.
.

[Figure]

Figure 7

[Figure]

Figure 8

**a** Hydrologic partioning

[Figure]

River inflow:
135.6 ± 119.0 mm d⁻¹

River outflow:
141.5 ± 132.4 mm d⁻¹

LGD: 10.3 ± 8.2 mm d⁻¹

Precipitation: ≈ 1 mm d⁻¹

Evaporation: 5.2 ± 4.7 mm d⁻¹

[Figure]

**b** DIN (μmol m⁻² d⁻¹)

1195.0
954.3
76.0  89.3
1000.5
1439.9

SOURCES    SINKS

**c** DIP (μmol m⁻² d⁻¹)

52.9
3.2
2.5
54.7
3.9

SOURCES    SINKS

GW input    Surface water input    Atmospheric deposit

Surface water outflow    Biological uptake    Other sinks